# Effects of Material Properties on Angular Distortion in Wire Arc Additive Manufacturing: Experimental and Computational Analyses

**DOI:** 10.3390/ma13061399

**Published:** 2020-03-19

**Authors:** Sang-Cheol Park, Hee-Seon Bang, Woo-Jae Seong

**Affiliations:** 1Industrial Application R&D Institute, Daewoo Shipbuilding & Marine Engineering Co., Ltd., 3370, Geoje-daero, Geoje-si, Gyeongsangnam-do 53302, Korea; gonausa2005@dsme.co.kr; 2Dept. of Welding and Joining Science Engineering College of Engineering, Chosun University, 309 Pilmun-daero, Dong-gu, Gwangju 61452, Korea; banghs@chosun.ac.kr

**Keywords:** wire arc additive manufacturing, angular distortion, thermomechanical property, bead-on-plate welding, finite element analysis

## Abstract

In wire arc additive manufacturing (AM), as in arc welding, arc heat thermally deforms substrates and articles. For industrial applications, deformation characteristics of various materials must be understood and appropriate materials and methods of reducing deformation must be devised. Therefore, angular distortions of different materials were investigated through bead-on-plate welding and finite element analysis. A model that simplifies temperature-dependent properties was developed to establish relationships between thermomechanical properties and angular distortion. A simplified model of temperature-dependent properties was used, and angular distortion characteristics were extensively investigated for different material properties and heat inputs. Coefficient of thermal expansion, density, and specific heat all notably affected angular distortion depending on heat input conditions. Results showed that during wire arc AM, flatness of both substrates and articles could vary depending on material properties, heat input, substrate thickness, and bead accumulation. Study findings can provide insight into deformation characteristics of new materials and how to mitigate thermal distortions.

## 1. Introduction

Considerable effort has been put into industrial applications of additive manufacturing (AM). In particular, AM shows great potential for application to large parts for heavy industry and shipping because it can shorten existing casting-centric processes. Thus, recent research has focused on increasing its productivity and developing various materials. In wire arc AM, metal wire is fused at its melting point by arc heat and then laminated layer by layer to form a desired shape. In wire arc AM, substrates and additive products are both deformed by repeated heating and cooling. Factors affecting AM-product flatness include interpass temperature and transient temperature distribution due to moving heat sources and shape-difference-induced nonuniform temperature distribution. In addition, substrate distortion can markedly affect manufactured-article flatness (Figure 1). Substrate thickness is limited in wire arc AM; therefore, optimal materials and substrate thicknesses must be determined efficiently by analyzing the effects of thickness and thermomechanical properties on thermal distortion of finished products. The latter is the subject of this study; the resulting data provide insight into deformation characteristics achieved when applying new materials to existing ones and to additive processing.

Mukherjee et al. [2] reported that AM-produced metal components were susceptible to thermal distortion, lack of fusion defects, and changes in composition-, all of which may be distorted depending on alloy thermophysical properties, part rigidity, and transient temperature fields. Furthermore, they presented strain and a thermal fluid-flow model to mitigate thermal distortion and analyzed the effects of power, scanning speed, Marangoni and Fourier numbers, and nondimensional peak temperature on thermal strain [3]. Xie et al. [4] showed that constraining force could explain and allow better understanding of distortion in metal AM. They demonstrated that temperature and cross-sectional area both played critical roles in determining constraining force.

AM-induced substrate and article thermal deformation follows the same principle as arc-welding-induced deformation. Thus, understanding fundamental principles of distortion is essential. Okerblom [5] identified a linear relationship between angular distortion and Q (heat input)/h^2^ (square of thickness) for bead-on-plate (BOP) welding when penetration was <0.6 h (thickness). Building on Okerblom’s work, Satoh and Terasaki [6] derived characteristics of shrinkage and angular distortion for each welding process through extensive BOP welding experiments. They found that in gas metal arc welding (GMAW), angular distortion of mild steel increased proportional to Q/h^2^ when Q/h^2^ < 2500 cal/cm^3^. They also compared welding distortions of various materials such as mild steel, aluminum alloy, and stainless steel using dimensionless heat input ($T_{0}^{*}=\left(\alpha_{0}/{c\rho \varepsilon }_{Y0}\right)\cdot Q/h^{2}$) and temperature parameters ($\theta_{M}^{*}$) representing decreasing material distortion resistance. Luo et al. [7] proposed a mechanism for generating welding distortion by inherent strain ($\varepsilon^{*}=-\varepsilon_{Y}/\beta =-\sigma_{Y}/\beta E$) and yielding temperature $\left(T_{1}=\sigma_{Y}/\beta E\alpha \right)$ at which stress just reached yield stress during heating. Mochizuki and Okano [8,9] proposed the parameter (b_m_/h)/(d_m_/h) as a mechanical melting zone, which was more accurate than the heat input parameter presented by Satoh and Terasaki [6], by considering heat-affected-zone width and height. Its usefulness was verified through analyses of various BOP welding processes; the subthreshold parameter demonstrated that reinforcement of the bead negligibly affected angular distortion. Their results demonstrated that size of the mechanical melting zone wherein the strain resistance was lost was a dominant parameter of weld angular distortion.

Finite element analysis is useful for quantifying relationships between angular distortion and material properties because it can examine effects of individual material properties on angular distortion while holding other properties constant, which is experimentally impossible. Vega et al. [10] found that specific heat, density, and thermal expansion coefficient all affected inherent plate deformation to a large degree; yield stress and Young’s modulus affected inherent deformation to some degree; thermal conductivity, Poisson’s ratio, and heat transfer coefficient all affected inherent deformation to a slight degree. They argued that strain hardening may be ignored for repeated heating lines. We adopted their methodology to select a finite element analysis model to discover effects of material properties on angular distortion. Guan et al. [11] investigated the influence of material properties on laser formation of sheet metals and found that bending angle was more sensitive to variation in yield strength than to variation in Young’s modulus. They also found that thermal expansion coefficient was crucial to laser formation. A small coefficient of heat conductivity resulted in a larger bending angle. Peak temperature in the heated zone was high for materials showing low specific heat and density so that sheets easily could be deformed. Zhu et al. [12] showed that thermal conductivity certainly affected distribution of transient temperature fields during welding. Yield stress and Young’s modulus considerably and slightly affected post-welding residual stress and distortion, respectively. Except for room-temperature yield stress, other room-temperature material properties reasonably predicted transient temperature fields, residual stress, and distortion. Yang et al. [13] studied effects of material strength and heat input on in-plain shrinkage and out-of-plane distortion of thin structures. They found that higher-strength steels showed less or similar out-of-plane distortions than lower-strength steels. With increasing welding heat input, shrinkage and distortion increased for both lower- and higher-strength materials. For the same heat input, shrinkage and distortion both decreased with increasing material thickness. Furthermore, research has been conducted to solve the problem by simply linearizing temperature-dependent nonlinear properties. Zhu et al. [12] suggested that simplified properties could be represented by a piecewise linear function of temperature for yield stress and room-temperature constants for all the other properties in computational weld simulations.

Although previous research has examined how material properties modulate distortion of AM-ed or welded parts, we seek to clarify effects of material properties on weld angular distortion. Therefore, we first conducted extensive BOP experiments on various materials, thicknesses, and heat input parameters. Then, we established a 3D thermal elastoplastic FEM (finite element method) model (hereafter: reference model) by applying simplified properties matching our experimental results. Finally, angular distortion characteristics were investigated for various materials under various heat input conditions, to allow determination of the effects of material properties on angular distortion.

## 2. Materials and Methods

Wire arc AM uses high-temperature arc heat, thereby inevitably introducing thermal contraction, which can be classified as transverse and longitudinal shrinkages according to direction. Out-of-plane deformation (i.e., transverse and longitudinal distortion) occurs when the shrinkage direction and product neutral axis are misaligned. Although thin materials may undergo buckling deformation, it is not so common in AM. This study only examines transverse angular distortion because it is easily identifiable and quantifiable. Previous studies [5,6,8,9] have reported that subthreshold angular distortion was proportional to heat input and inversely proportional to the square of thickness. Herein, we define this relationship as a heat input parameter, representing electric energy divided by traveling speed, to evaluate angular distortion. The parameter is given by Equation (1) as follows:
(1)$$\mathrm{Heat}\;\mathrm{input}\;\mathrm{parameter}:\;\frac{Q_{\mathrm{eff}}}{h^{2}}=\eta \cdot \frac{\mathrm{VI}}{u}\cdot \frac{1}{h^{2}}\left(J/{\mathrm{mm}}^{3}\right)$$
where Q_eff_ is the effective heat input transferred to the base metal (or substrate), η arc efficiency, V is voltage (V), I is welding current (A), u welding speed (mm/s), and h is substrate thickness (mm).

First, BOP welding experiments with various materials were performed based on this parameter. Then, the reference material and its simplified properties were set, and relationships between angular distortion and material properties were investigated via finite element analysis. Finally, the results of the analysis and the experiments were compared.

### 2.1. Materials

Materials commonly used in mechanical parts and structures of ships and offshore platforms, including carbon steels such as B, AH32, EH40-TM, A500, and HY80, and low-temperature steels such as 9% Ni steel, stainless steel (SUS316L), and aluminum alloy steel (AL6061-O) were used in the experiments. Chemical compositions and mechanical properties of the materials are summarized in Table 1.

### 2.2. BOP Welding Experiments

BOP welding experiments were performed on various material plates. Figure 2 shows the specimen size. To consider only transverse angular distortion, specimens were wider (1000 mm) than they were long (400 mm). The welding area shadow divided specimens into thirds. Plates were welded at two areas under different heat input conditions. Temporary pieces were tack welded at the beginning and end of the weld to obtain a stable initial arc and avoid transient thermal conduction. GMAW was applied to AL6061-O with 99% Ar as a shielding gas. Most welding experiments were performed by flux-cored arc welding (FCAW) with CO_2_ 99% as the shield gas. All the wires were 1.2 mm in diameter. Table 2 lists the welding conditions for all the BOP experiments.

Various heat inputs were imposed by fixing welding current voltage and adjusting travel speed according to parameters shown in Table 3. An inverter-type welding machine and a smart feeder were used to easily control current and voltage, thereby enabling identical heat input conditions to be achieved during welding. The distance between the specimen and torch (i.e., the standoff distance) was maintained between 19.5 and 20 mm. Figure 3 shows welded specimens used to measure angular distortion.

We used a maximum of 16 passes and a minimum of 5 passes to deposit small and large welds, respectively (Table 3). For each pass, angular distortion was measured using a digital angle gauge posed on metal cubes, as shown in Figure 4. Interpass temperature was maintained below 40 °C. Average angular distortion was adopted because the distortion angle is constant for the same heat input, material, and specimen thickness regardless of the number of passes [1].

### 2.3. Thermal Elastoplastic Analysis

To determine how material properties affected angular distortion, we examined density, specific heat, thermal conductivity, thermal expansion coefficient, elastic modulus, and yield strength. Properties of each material according to the literature are listed in Table 4. The material properties vary with temperature; therefore, each property has a corresponding range.

AH32 was selected as a reference material to quantitatively compare how much different material properties changed angular distortion. In addition, we set material properties by a reference model proposed by Zhu et al. [12]. A reference model is a method of simplifying temperature-dependent properties showing different tendencies for different materials. It is introduced herein because quantitatively comparing materials is easy. We used room-temperature material density and average specific heat, thermal conductivity, and thermal expansion coefficients over the entire temperature history. The room temperature yield strength and elasticity modulus were used for temperature up to 100 °C, which then decreased linearly to 5% of room temperature values at 2/3 of the melting point temperature. Above this temperature, the values remained constant. Elevated-temperature AH32 material properties are shown in Figure 5a,c, and corresponding reference models are shown in Figure 5b,d. Analysis was performed by varying targeted property variables while fixing all the other property variables.

In addition to material property setup, heat sources were established for computation. Uniform body heat flux, calculated using Equation (2), was applied as thermal load, as shown in Figure 6.
(2)$$q_{M}=\frac{Q_{\mathrm{eff}}}{A_{W}L_{w}}\left(\frac{W}{{\mathrm{mm}}^{3}}\right)$$
where η = 0.8, A_W_ is weld-bead cross-sectional area (mm^2^), and L_w_ = B_w_ is bead width (mm).

Figure 7a shows the result of the proposed moving heat source. Convection cooling was used as a boundary condition and the convective coefficient is shown in Figure 5b. Ambient temperature was set at 20 °C. A symmetric half-model weld line was adopted, and the bead already had been modeled prior to welding. To prevent rigid-body movement, two corners were fixed in the X-direction, and a rigid surface was used below the plate to prevent gravitational movement in the Z-direction (Figure 7b). Computational analysis was performed using ABAQUS 2018 (Dassault Systemes Simulia Corp., Johnston, RI, USA). DC3D8 (a provided element) and C3D8R were used for heat transfer and elastoplastic analyses, respectively. To implement the moving volumetric heat source, we used the Dflux User Subroutine in Fortran language. The number of elements was determined by a meshing test.

Thermal elastoplastic analyses were performed using both reference and temperature-dependent models for 12-mm-thick AH32 BOP welding. Heat input conditions listed in Table 5 were identical to welding conditions shown in Table 3. Angular distortion was extracted from displacements at each node of the analysis results.

## 3. Results

### 3.1. Experimental Results

We examined effects of heat input parameter Q_eff_/h^2^ (calculated according to Equation (1)) on angular distortion of each material. Arc efficiency (GMAW) was set to 0.8 [18].

As shown in Figure 8, all the materials showed similar angular distortion trends; that is, angular distortions increased proportional to the increase in heat input parameters up to the threshold; above this, it decreased nonlinearly. Table 6 quantitatively summarizes some of the data shown in Figure 8. EH40-TM (18t) and A500 (20t) carbon steels both were tested in a low heat input parameter range; therefore, no threshold was observed. Conversely, materials B, AH32, HY80, 9% Ni steel, and stainless steel showed thresholds at specific heat input parameters.

Carbon steels AH32, B, and HY80 (12t) (yield strengths = 348, 295, and 626 MPa) showed thresholds in the ranges 12–13, 9–10, and 13–14 J/mm^3^, respectively Clearly, angular distortion decreased with increasing yield strength at heat input parameters below 10 J/mm^3^ while angular distortion increased with increasing yield strength at heat input parameters above 13 J/mm^3^ (columns Q1 and Q4 in Table 6). In addition, although the slope of subthreshold angular distortion decreased slightly, the threshold itself increased with increasing yield strength.

Nine percent Ni steel showed the highest threshold in the range 17–18 J/mm^3^. Because 9% Ni steel showed the smallest subthreshold slope, it showed the least angular distortion of all the materials. Stainless steel, on the other hand, showed a threshold in the range 14–15 J/mm^3^ and the most angular distortion of all the materials. The aluminum alloy showed the lowest threshold in the range 5–6 J/mm^3^ and appeared to follow a trend similar to that for stainless steel. Low heat input parameters showed the greatest angular distortions.

### 3.2. Thermal Elastoplastic Analysis Results

To verify the reference model, results obtained using the reference and temperature-dependent models were compared with experimental results. Figure 9 compares all the results obtained for AH32. The results obtained for the reference and temperature-dependent models both showed the same trend as the experimental results. Below the threshold, error between the reference model and experimental results was 1–17%, and that between the temperature-dependent model and experimental results was 1–31%. Above the threshold, in the high heat input zone, results obtained using the temperature-dependent model were more consistent with experimental data (within 1.3%) than those obtained using the reference model were (within 19%). Although the reference model had simplified temperature-dependent material properties, results were reasonable under all heat input conditions. However, analysis results were higher than experimental ones owing to arc-efficiency accuracy because arc efficiency had conservatively been set slightly higher. Results demonstrated that reference-model-based material properties were available instead of temperature-dependent properties for quantitative comparison of properties.

## 4. Discussion

The effects of material properties on angular distortion were examined using the reference model under various heat input conditions (Q1, Q2, Q3, and Q4).

To compare angular distortions of various materials, the properties of AH32 material were used as references, and the concept of property ratio (the ratio of the difference between AH32 and target material properties to the AH32 property) was introduced, as shown in Figure 10. For example, if the target material is AH32 and the target property is yield strength, yield strength property ratio = 1.0. For comparison, angular distortion was also defined as the difference between AH32 and target material properties. Material property ratio and heat input were calculated by computational analysis while adjusting only target properties and fixing all other properties.

### 4.1. Effect of Density on Angular Distortion

The effect of density on angular distortion differed under various heat input conditions (Figure 11). Under subthreshold heat input conditions (Q1, Q2, and Q3), angular distortion decreased linearly with increasing density ratio above 0.64. However, angular distortion increased with increasing density under the suprathreshold heat input condition (Q4).

Angular distortion obtained at density ratio = 1 was identical to that calculated for AH32 12t steel.

Heat capacity, the product of specific heat and density, is the ability to store heat energy rather than conduct it. Thermal diffusivity is defined as:(3)$$\kappa =\frac{\lambda }{\rho C}\;$$
where ρ density, C is specific heat, and κ is thermal conductivity.

Thermal diffusivity represents rate of heat transfer and is proportional to thermal conductivity. According to Equation (3), the higher the material density, the slower the heat will be transferred. This is because the material can store more energy; therefore, heat is concentrated rather than dissipated.

As shown in Table 7, the area under the isotherm at 725 °C decreased with increasing density ratio for heat input Q1. The peak temperature field is relevant to plastic strain distribution; therefore, the plastic area also decreased with increasing density ratio and thereby decreased angular distortion. The peak temperature field here refers to the distribution of the highest temperatures experienced inside the material during the heating and cooling process. Although area under the isotherm at 725 °C decreased with increasing density ratio, for heat input Q4, temperature and plastic-strain gradients both were steeper at the higher density ratio along the thickness direction. Therefore, the angular distortion increased; this implied that heat was concentrated at the top of the plate near the heat source. In addition, when the heat input conditions (Q2, Q3) were less than the threshold (0.64), the angular distortion decreased rapidly when the density ratio decreased. This is because the difference in the plastic strain in the thickness direction is small due to the high thermal diffusivity. Results showed that the temperature gradient through the material thickness depends on density, and the tendency toward angular distortion can change with different heat inputs.

### 4.2. Effect of Specific Heat on Angular Distortion

Heat capacity, the product of specific heat and material density, is the ability of a material to store heat energy rather than conduct it. Therefore, the higher the material specific heat, the smaller the heating area. Because specific heat and material density show the same heat distribution pattern, plastic strain distribution is similar. Effect of specific heat on angular distortion depended on various heat input conditions (Figure 12). In subthreshold heat input ranges (Q1, Q2, and Q3), the compressive-plastic-strain area at the top of the plate near the heat source was reduced, resulting in decreased angular distortion with increasing specific heat. Under the suprathreshold heat input condition (Q4), on the other hand, angular distortion increased with increasing specific heat. For high heat inputs, the greater the specific heat, the more heat concentrated at the top of the plate near the heat source. Thus, angular distortion increased with increasing specific heat.

### 4.3. Effect of Thermal Conductivity on Angular Distortion

Effect of thermal conductivity on angular distortion depended on various heat input conditions (Figure 13). Thermal conductivity of aluminum alloy is in the range 150–175 W/m·°C, which is 4.4 times that of carbon steel. Therefore, the thermal conductivity ratio range was extended. The results indicated that thermal conductivity affected angular distortion less than density or specific heat did. The lower the heat input, the smaller the difference. The lowest heat input condition (Q1) negligibly affected angular distortion. Under subthreshold heat input conditions, angular distortion tended to decrease with increasing thermal conductivity because thermal conductivity is proportional to thermal diffusivity, as opposed to density and specific heat, suggesting that the plastic area was reduced, thereby decreasing angular distortion. In contrast, under the suprathreshold heat input condition (Q4), angular distortion increased with increasing thermal conductivity up to a certain point. However, compared to other thermal properties, the effect of thermal conductivity on angular distortion was still very small (Figure 13).

According to the heat conduction (Fourier–Biot) equation, the closer a material is to the steady state, the greater the effect of thermal conductivity on temperature and the closer a material is to the transient state, the greater effects of density and specific heat on temperature. In welding, temperature changes in a very wide range very quickly (transient state), so density and specific heat dominate heat conduction. Therefore, density and specific heat affected thermal distortion more than thermal conductivity did.

### 4.4. Effect of Thermal Expansion Coefficient on Angular Distortion

Effect of the thermal expansion coefficient on angular distortion depended on various heat input conditions (Figure 14). Angular distortion increased linearly with increasing thermal expansion coefficient ratio. The more expansion increased with increasing temperature during heating, the faster yield stress was reached. Compressive plastic strain increased and distribution widened, thereby increasing distortion. The suprathreshold heat input condition affected angular distortion less than subthreshold heat input conditions did, suggesting that thermal-expansion-induced compressive plastic strain simultaneously increased at the top and bottom of the plate, thereby reducing angular distortion.

### 4.5. Effect of Elastic Modulus on Angular Distortion

Effect of elastic modulus on angular distortion depended on various heat input conditions (Figure 15). Under all the heat input conditions except for Q3 and Q4, angular distortion increased with increasing elastic modulus. Strain at which plasticity begins is defined as yield strength divided by elasticity modulus. In other words, with increasing elastic modulus, plasticity begins at lower strains, meaning that plastic deformation occurs easily. However, under the suprathreshold high heat input condition (Q4), angular distortion decreased with increasing elastic modulus. For high heat inputs, the temperature gradient is small in the thickness direction, thereby relatively evenly distributing plastic strain. Nevertheless, plastic strain at the top of the plate was greater than that at the bottom. Under this condition, the smaller the elasticity modulus, the lower the plastic strain from the bottom of the plate; therefore, angular distortion increased because it results from the difference in plastic strain at the top and bottom of the plate.

### 4.6. Effect of Yield Strength on Angular Distortion

Effect of yield strength on angular distortion depended on various heat input conditions (Figure 16). Under the lowest heat input condition (Q1), angular distortion decreased almost linearly with increasing yield strength. Although yield strength tends to behave oppositely to elasticity modulus, internal mechanisms for both are almost identical. For low heat inputs, plastic strain was mainly distributed at the top of the plate near the heat source. Higher yield strength reduces plasticity, thereby decreasing angular distortion. In contrast, under the highest suprathreshold heat input condition (Q4), angular distortion increased linearly with increasing yield strength. Because temperature gradient was small in the thickness direction for high heat inputs, high yield strength reduced plastic deformation at the bottom of the plate, thereby increasing angular distortion. Under conditions Q2 and Q3, angular distortion increased and decreased with increasing yield strength, respectively, as shown in Figure 8.

### 4.7. Property-angular distortion ratio

We examined effects of thermal and mechanical properties on angular distortion under various heat input conditions and corresponding causes to quantify effects of material properties on angular distortion. Ratios were linearly calculated based on the definition of property angular distortion ratio (Figure 10) and are listed in Table 8.

According to property angular distortion ratios, order of effects of various material properties on angular distortion can be arranged (Figure 17). Density and specific heat substantially influenced angular distortion under all heat input conditions. Thermal conductivity slightly affected angular distortion under all heat input conditions.

Although the thermal expansion coefficient largely affected angular distortion under subthreshold heat input conditions (Q1, Q2, and Q3), it had less of an effect on angular distortion under the suprathreshold heat input condition (Q4). For subthreshold heat input conditions, effects of thermal expansion coefficient and thermal conductivity on angular distortion both increased with increasing heat input.

Yield strength and elastic modulus both only slightly affected angular distortion under subthreshold heat input conditions, but they both affected angular distortion more than thermal expansion coefficient did under the suprathreshold heat input condition. Furthermore, the trends of increasing and decreasing angular distortion were opposite under low heat and high heat conditions.

The results indicated that when high heat input is used for high productivity in the arc wire AM process, the material properties affecting the distortion differ from those of the low heat input. For manufacturing large parts or structures, the range below Q2 appears to be controllable. In this range, the influence of specific heat, density and coefficient of thermal expansion on deformation is considerable. The higher the specific heat and density, and the smaller the coefficient of thermal expansion, the lower the amount of distortion. When applying new materials, these data can be used to predict the angular distortion and the heat input and substrate thickness can be adjusted to find optimal AM conditions.

### 4.8. Angular Distortion Characteristics of Various Materials

Considering differences between material properties of AH32 steel and various materials and between property-angular distortion ratios, the characteristics of angular distortions for various materials were discussed compared to AH32 steel.

Carbon steels in this study have different yield strength, and analysis results demonstrated that angular distortion decreased with increasing yield strength under subthreshold heat input condition (Q2), as experimentally investigated (Figure 8). Analysis results also demonstrated the opposite trend where angular distortion increased with increasing yield strength for suprathreshold heat input condition (Q4).

Stainless steel showed specific heat, thermal expansion, thermal conductivity, and yield strength on average 0.77, 1.5, 0.6, and 0.74 times those of carbon steel (AH32), respectively (Table 4). Accounting for property-angular distortion ratios and properties different from those of AH32 steel (Table 8), the effect of thermal expansion on angular distortion was the greatest with increasing angular distortion followed by specific heat under subthreshold heat input condition (Q2). Under the suprathreshold heat input condition (Q4), the effect of specific heat on angular distortion was the greatest with decreasing angular distortion followed by thermal expansion coefficient with increasing angular distortion. Although materials showing lower yield strengths than AH32 steel could decrease angular distortion under the suprathreshold heat input condition, magnitudes decrease in angular distortion were lower than those induced by the thermal expansion coefficient and specific heat.

The 9% Ni steel showed considerably different yield strength (2 times) and thermal conductivity (0.83 times) than AH32 steel (Table 4). Owing to these different material properties, the effect of yield strength on angular distortion was the greatest with deceasing angular distortion under subthreshold heat input condition (Q2) and with increasing angular distortion under the suprathreshold heat input condition (Q4), which supports the experimental result that angular distortion of 9% Ni steel was smaller than those of carbon steels under subthreshold heat input conditions and larger under the suprathreshold heat input condition.

Compared to AH32 steel properties, all the aluminum alloy properties affected angular distortion. Aluminum alloy showed density, specific heat, thermal conductivity, Young’s modulus, and yield strength 0.34, 1.45, 4.43, 0.33, and 0.32 times those of AH32 steel, respectively. (Table 4) Owing to different material properties and property-angular distortion ratios, effects of density and thermal expansion coefficient on angular distortion were most notable with increasing angular distortion followed by specific heat and Young’s modulus with deceasing angular distortion. Despite aluminum alloy showing thermal conductivity much higher than that of carbon steel, its lowest property-angular distortion ratio meant that its thermal conductivity only slightly affected angular distortion.

Results suggest that base-metal angular distortion characteristics can be estimated based on substrate thickness, heat input, and material properties of the substrate.

### 4.9. Comparison of Current and Previous Study Results

Effect of elastic modulus on angular distortion in this study was opposite to those shown in previous line heating studies of Vega et al. [10] and Guan et al. [11] During line heating, angular distortion decreased when the elastic modulus was large. For carbon steel, the maximum temperature used during line heating is usually around 800 °C. Thus, peak temperature is not high compared to welding temperature. Furthermore, Guan et al. did not apply temperature-dependent elastic moduli to FEM analyses, meaning that elastic moduli were applied equally despite decreasing owing to the difference in temperature around the flame. Moreover, the welded part showed a vast plastic region and temperature deviation compared to those generated by line heating. Therefore, it is difficult to compare their results with ours.

## 5. Conclusions

We investigated the angular distortion characteristics of various materials with bead-on-plate welding experiments. We extended our findings by using thermal elastoplastic analysis using the reference model. We analyzed the effects of material properties on the angular distortion under various heat input conditions. The main conclusion of this study can be summarized as follows:All the materials showed similar angular distortion trends; that is, angular distortion proportionally increased with increasing heat input parameter up to the threshold, above which it nonlinearly decreased.Materials B, AH32, HY80, 9% Ni steel, and stainless steel showed thresholds at specific heat input parameters, which indicates that there were specific thickness and heat input conditions that maximize angular distortion.Density and specific heat substantially influenced angular distortion under all heat input conditions. Thermal conductivity slightly affected angular distortion under all heat input conditions.Although the thermal expansion coefficient largely affected angular distortion under subthreshold heat input conditions, it had less effect on angular distortion under the suprathreshold heat input condition.Although yield strength and elastic modulus both only slightly affected angular distortion under subthreshold heat input conditions, they both affected angular distortion more than the thermal expansion coefficient did under the suprathreshold heat input condition.The effect of material properties on angular distortion depends on the heat input. This is because heat input affects the difference in temperature and plastic strain in the thickness direction of the substrate.

## Figures and Tables

**Figure 1 materials-13-01399-f001:**
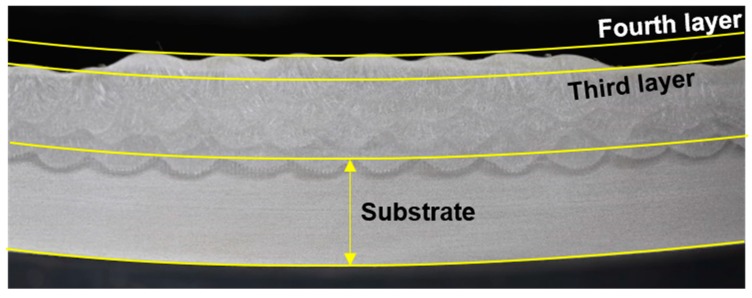
Cross-section of multilayer beads demonstrating thermal distortion [1].

**Figure 2 materials-13-01399-f002:**
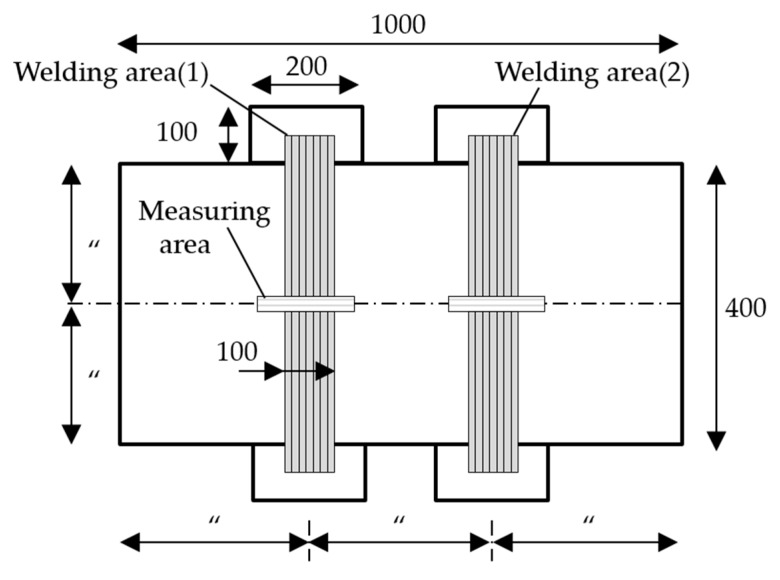
Schematic of welding specimen. Measurements are in mm.

**Figure 3 materials-13-01399-f003:**
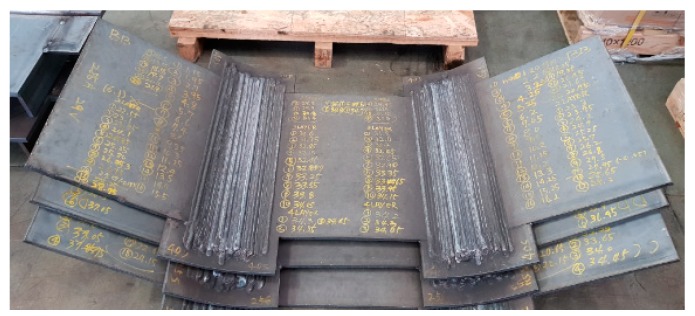
Welded specimens used to measure angular distortion.

**Figure 4 materials-13-01399-f004:**
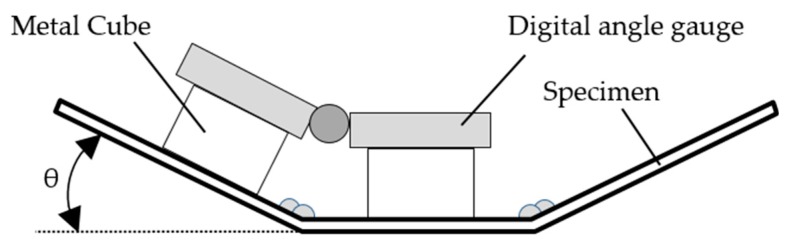
Schematic of angular distortion measurement.

**Figure 5 materials-13-01399-f005:**
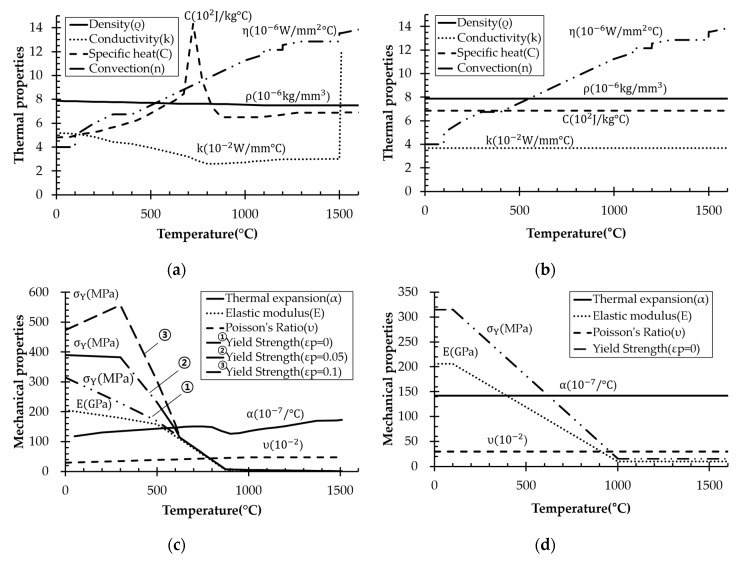
Thermal properties (**a**) and (**b**) and corresponding mechanical properties (**c**) and (**d**) of temperature-dependent and reference models for AH32 steel.

**Figure 6 materials-13-01399-f006:**
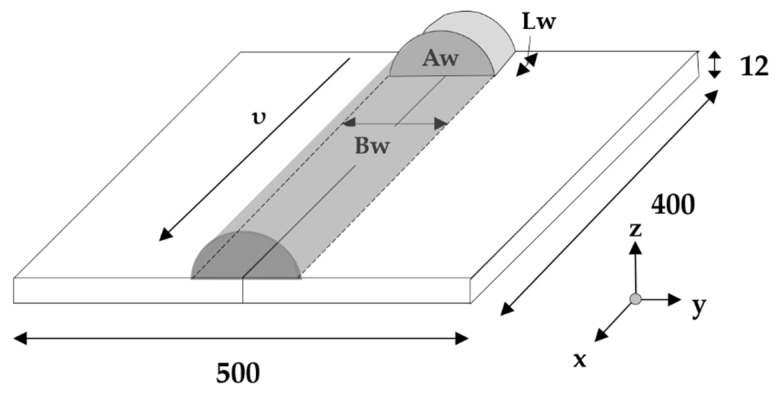
Uniform body heat flux application and definition.

**Figure 7 materials-13-01399-f007:**
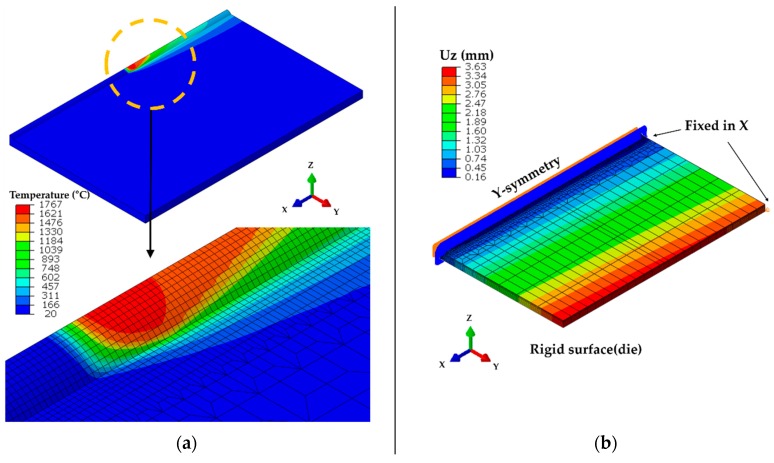
FEM analysis model: (**a**) Analysis of heat transfer under a moving heat source; (**b**) Boundary condition for thermal elastoplastic analysis.

**Figure 8 materials-13-01399-f008:**
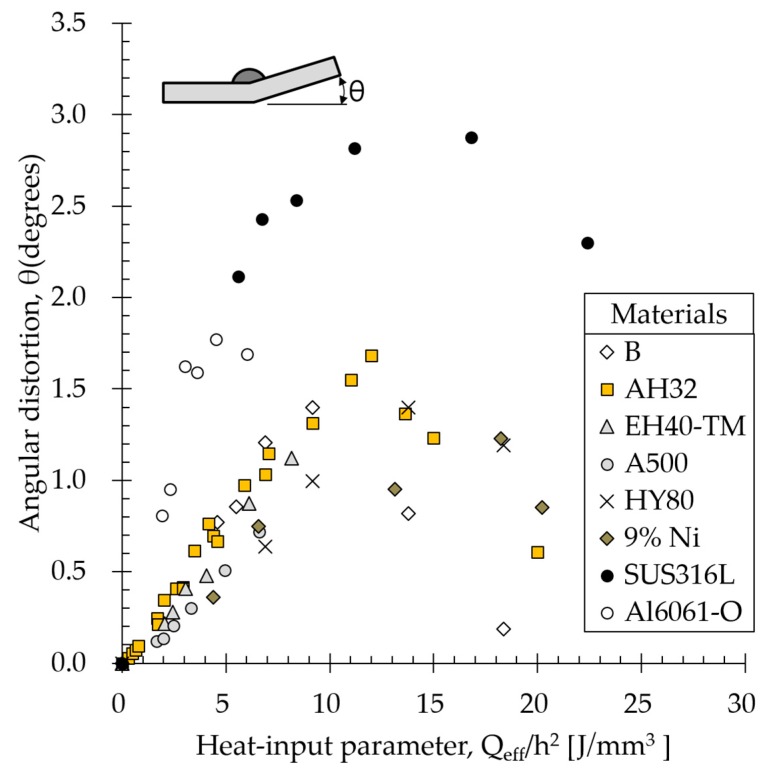
Experimental results: heat input parameter vs. measured angular distortion.

**Figure 9 materials-13-01399-f009:**
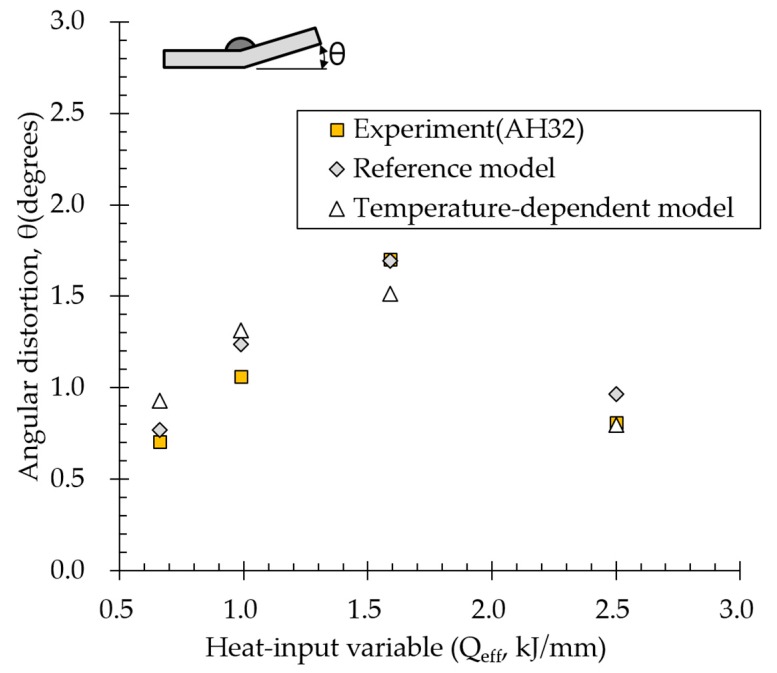
Angular distortion measured by experiment and two proposed models.

**Figure 10 materials-13-01399-f010:**
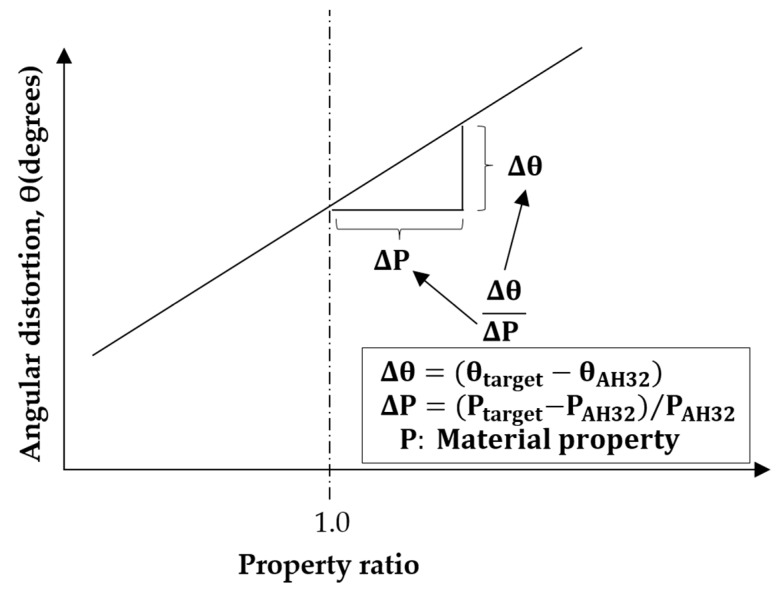
Definition of material property ratio.

**Figure 11 materials-13-01399-f011:**
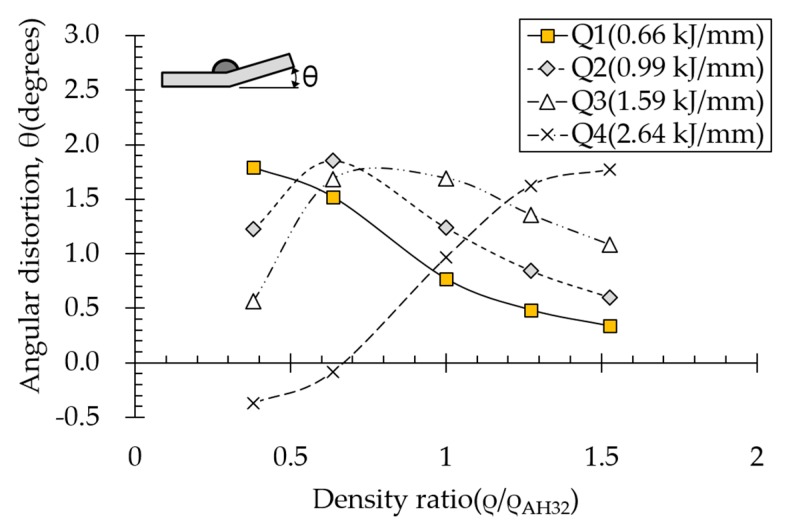
Effect of density on angular distortion.

**Figure 12 materials-13-01399-f012:**
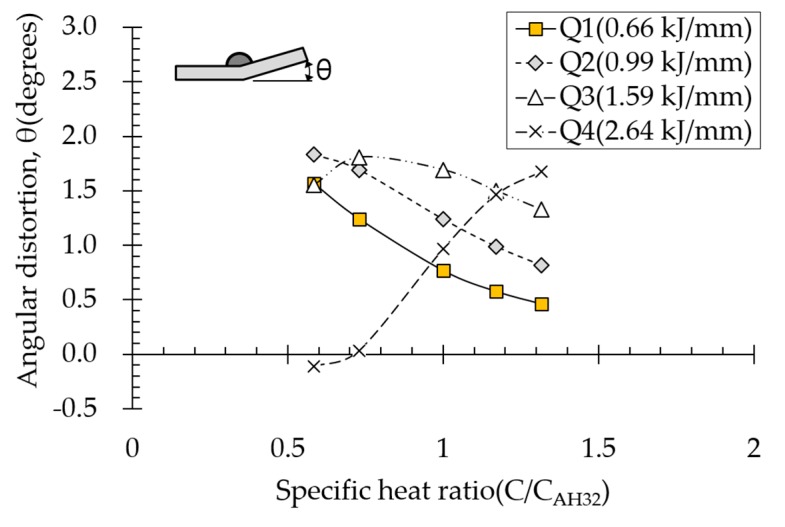
Effect of specific heat on angular distortion.

**Figure 13 materials-13-01399-f013:**
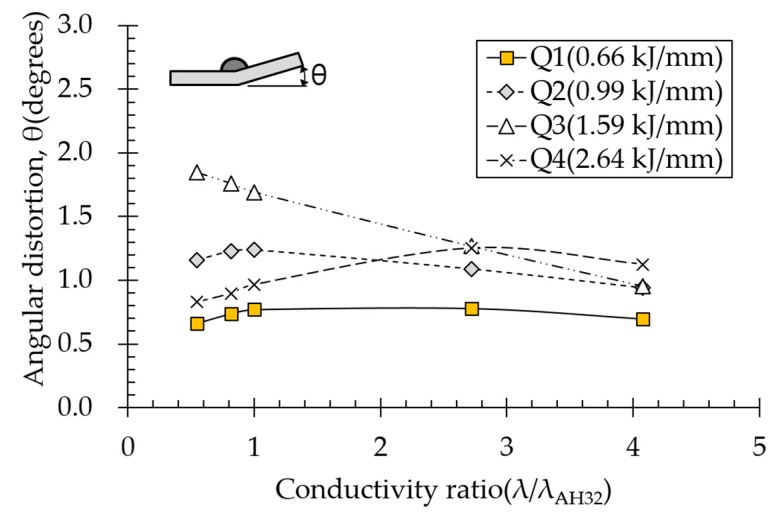
Effect of thermal conductivity on angular distortion.

**Figure 14 materials-13-01399-f014:**
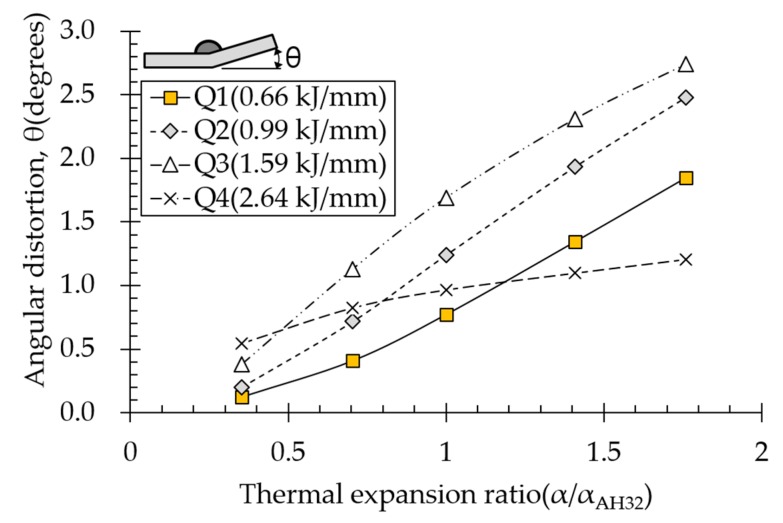
Effect of thermal expansion on angular distortion.

**Figure 15 materials-13-01399-f015:**
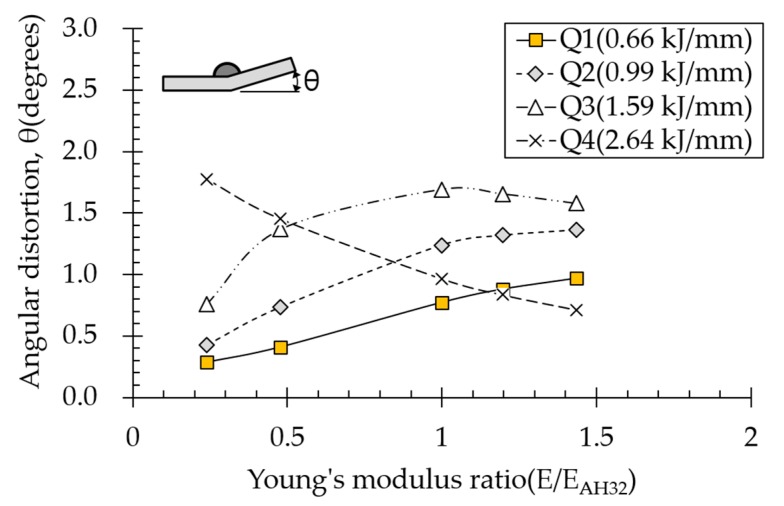
Effect of Young’s modulus on angular distortion.

**Figure 16 materials-13-01399-f016:**
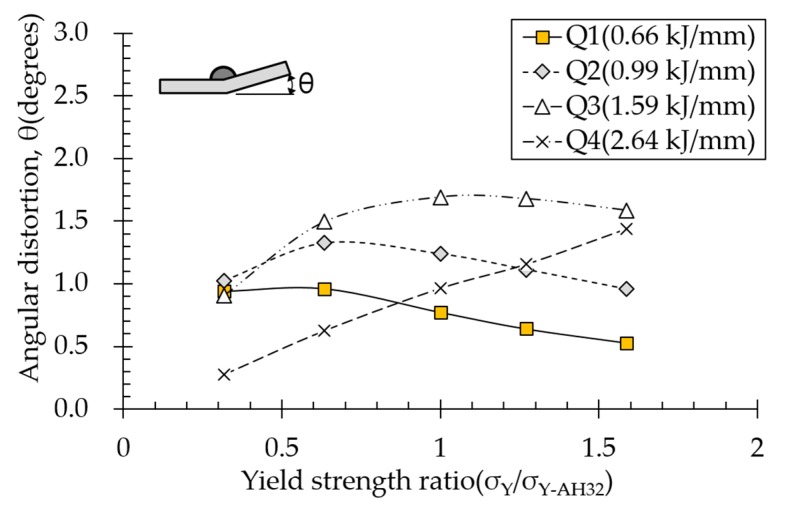
Effect of yield strength on angular distortion.

**Figure 17 materials-13-01399-f017:**
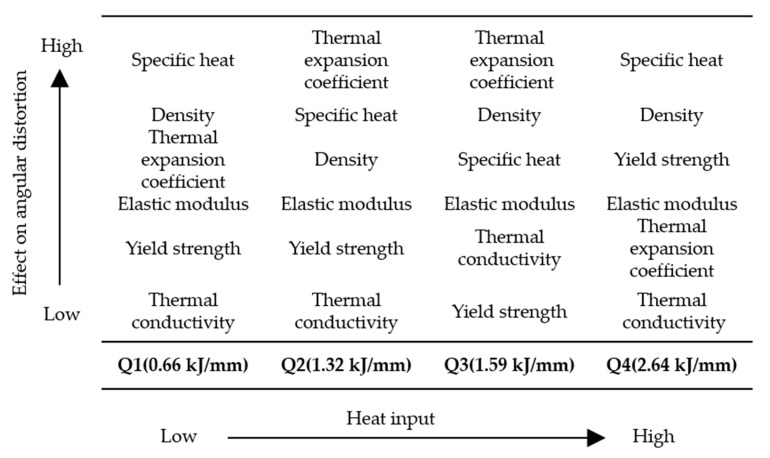
Order of effects of material properties on angular distortion.

**Table 1 materials-13-01399-t001:** Chemical compositions and mechanical properties of various materials.

Material	Chemical Composition (mass %)					Mechanical Properties		
	C	Si	Mn	Ni	Cr	Yield Strength (MPa)	Tensile Strength (MPa)	Elongation (%)
B	0.12	0.19	0.94	0.00	0.02	295	442	29
AH32 [1]	0.16	0.24	1.12	0.04	0.03	348	495	30
EH40-TM	0.08	0.19	1.52	0.35	0.02	504	558	22
A500	0.08	0.31	1.51	0.02	0.16	559	666	17
HY80	0.14	0.25	0.25	3.08	1.59	626	768	27
9% Ni	0.56	0.25	0.61	9.00	0.02	684	721	28
SUS316L	0.02	0.50	1.26	10.1	16.7	285	588	58
AL6061-O	According to KS D6759 [14]					110	145	16

**Table 2 materials-13-01399-t002:** Welding conditions for BOP experiments.

Material	Thickness (h, mm)	Welding Process	Polarity	Shield Gas
B	12	FCAW ^1^	DCEN ^3^	99% CO_2_
AH32 [1]	11.5, 12, 15 19.5, 28, 45	FCAW	DCEN	99% CO_2_
EH40-TM	18	FCAW	DCEN	99% CO_2_
A500	20	FCAW	DCEN	99% CO_2_
HY80	12, 15	FCAW	DCEN	99% CO_2_
9% Ni	15	FCAW	DCEN	99% CO_2_
SUS316L	10	FCAW	DCEN	99% CO_2_
AL6061-O	10,15	Pulse-GMAW ^2^	DCEN	99% Ar

^1^ Flux-cored arc welding; ^2^ Gas metal arc welding; ^3^ Direct current electrode negative.

**Table 3 materials-13-01399-t003:** Heat input conditions for BOP experiments.

Material	Current (A)	Voltage (V)	Speed (cm/min)–(Number of Pass per Layer)					
B	285	29	60–(16)	50–(13)	40–(13)	30–(11)	20–(8)	15–(7)
AH32 [1]	285	29	60	40	30	25	-	-
EH40-TM	285	29	60–(12)	50–(12)	40–(7)	30–(7)	20–(8)	15–(8)
A500	285	29	60–(13)	50–(12)	40–(10)	30–(10)	20–(8)	15–(8)
HY80	285	29	–	–	40–(9)	30–(9)	20–(8)	15–(7)
9% Ni	220	28	–	30–(6)	20–(7)	10–(7)	7.2–(5)	6.5–(5)
SUS316L	250	28	60–(7)	50–(7)	40–(6)	30–(6)	20–(6)	15–(6)
AL6061-O	208	26	60–(27)	50–(25)	40–(11)	30–(10)	–	–

**Table 4 materials-13-01399-t004:** Thermomechanical property ranges of various materials.

Material Properties	Carbon Steels (B, AH32, EH40-TM, A500, HY80)	9% Ni [15]	SUS316L [16]	AL6061-O [17]
Density (ρ, kg/m^3^)	7490–7850	7230–7850	8020	2707
Specific heat (C, J/kg °C)	480–1430	450–1030	440–620	800–1040
Conductivity (λ, W/m °C)	26–52	25–36	14–33	147–179
Thermal expansion (α, ×10^−5^/°C)	1.17–1.73	1.07–1.86	1.6–2.7	2.45
Elastic modulus (E, GPa)	209	197	194	70
Yield strength (σ_Y_, MPa)	295–626	684	258	110

**Table 5 materials-13-01399-t005:** Heat input conditions for welding analysis.

Heat Input Condition (Q_eff_, kJ/mm)		Efficiency (η)	Current (A)	Voltage (V)	Velocity (mm/s)
Q1	0.66	0.8	285	29	10.0
Q2	0.99	0.8	285	29	6.67
Q3	1.59	0.8	285	29	4.17
Q4	2.64	0.8	285	29	2.5

**Table 6 materials-13-01399-t006:** Angular distortion characteristics of various materials.

Material	Threshold	Angular Distortion at Threshold (θ)	Subthreshold Slope $\left(\Delta \theta /\Delta (\frac{Q_{eff}}{h^{2}}\right))$	Angular Distortion (θ)	
				Q1 at 12t (0.99 kJ/mm)	Q4 at 12t (2.64 kJ/mm)
B	9–10	1.4	0.16	1.06	0.19
AH32	12–13	1.63	0.13	0.92	0.81
A500	–	–	0.11	0.69	-
HY80	13–14	1.41	0.10	0.70	1.38
9% Ni	17–18	1.35	0.07	0.65	0.61
SUS316L	14–15	2.81	0.29	2.40	1.20
AL6061-O ^1^	5–6	1.77	0.81	1.71 ^1^	-

^1^ Angular distortion of 12-mm-thick AL6061-O under Q1 condition (0.66 kJ/mm).

**Table 7 materials-13-01399-t007:** Angular distortions calculated from heat transfer and thermal elastoplastic analyses compared according to density ratio.

Heat Input	Results	$\rho /\rho_{AH32}=0.64$	$\rho /\rho_{AH32}=1$	$\rho /\rho_{AH32}=1.27$
Q1	H1 ^1^	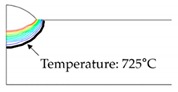	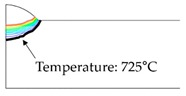	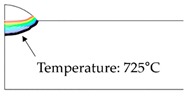
	P1 ^2^	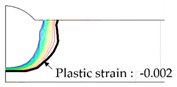	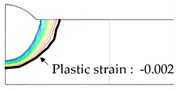	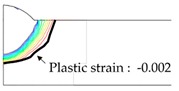
Q4	H4 ^1^	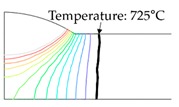	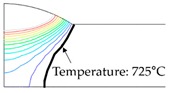	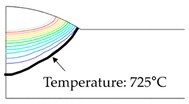
	P4 ^2^	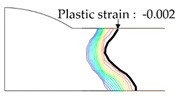	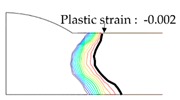	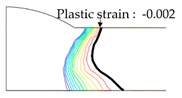

^1^ Results calculated from heat transfer analysis; ^2^ Results calculated from thermal elastoplastic analysis.

**Table 8 materials-13-01399-t008:** Property angular distortion ratios obtained under various heat input conditions.

Material Property	Q1 (0.66 kJ/mm)	Q2 (1.32 kJ/mm)	Q3 (1.59 kJ/mm)	Q4 (2.64 kJ/mm)
Density	−1.36 ^1^	−1.42	−1.16	2.06
Specific heat	−1.52	−1.45	−0.81	2.67
Thermal conductivity	0	−0.07	−0.25	−0.09
Thermal expansion coefficient	1.25	1.64	1.67	0.45
Elastic modulus	0.6	0.81	−0.26	−0.89
Yield strength	−0.36	−0.39	−0.18	0.9

^1^ Note that the negative sign (−) means that the angular distortion decreases.

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
