# Peer review of "Effects of Material Properties on Angular Distortion in Wire Arc Additive Manufacturing: Experimental and Computational Analyses"

_materials, 2020, doi:10.3390/ma13061399_

Round 1
Reviewer 1 Report
On behalf of materials-741968: “Effects of material properties on angular distortion in wire arc additive manufacturing: Experimental and computational analyses”
- Why have you discarded Q3? Do results for Q3 fall between those obtained for Q2 and Q4? I don't really see the point of removing this condition from the discussion.
- Behavior of response (angular distortion) is individually analyzed for each property. Have you checked cross-influence between parameters?
- I recommend you to introduce “Conclusions” section to clarify the usefulness of your findings, since it is not easy to figure out how practical applications can be derived from your work.
- In Table 5, Heat Imput Condition is expressed in kJ / mm, but in Fig. 9, it seems to be expressed in kJ / m. Is it a typo? Which is correct?
- There are several typos across the text, specially unexpected spaces at the beginning of paragraphs. Please check the text carefully.
Author Response
- Point 1:
Why have you discarded Q3? Do results for Q3 fall between those obtained for Q2 and Q4? I don't really see the point of removing this condition from the discussion.
Response 1:
Thanks for your comment. Simulation was performed under Q3 conditions. However, for understanding and readability, only four conditions were indicated in the manuscript by replacing Q4 with Q3 and Q5 with Q4.
- Point 2:
Behavior of response (angular distortion) is individually analyzed for each property. Have you checked cross-influence between parameters?
Response 2:
Finite element method we used is based on the continuum mechanics where all material properties are independent. There is no need to check the cross-effect between material properties.
- Point 3:
I recommend you to introduce “Conclusions” section to clarify the usefulness of your findings, since it is not easy to figure out how practical applications can be derived from your work.
Response 3:
As per your recommendation, a separate chapter 5 for the conclusion was made in the revised paper. The following statement was added.
- Conclusions
We investigated the angular distortion characteristics of various materials with bead-on-plate welding experiments. We extended our findings by using thermal elastoplastic analysis using the reference model. We analyzed the effects of material properties on the angular distortion under various heat input conditions. The main conclusion of this study can be summarized as follows:
- All the materials showed similar angular distortion trends; that is, angular distortion proportionally increased with increasing heat input parameter up to the threshold, above which it nonlinearly decreased.
- Materials B, AH32, HY80, 9% Ni steel, and stainless steel showed thresholds at specific heat input parameters, which indicates that there were specific thickness and heat input conditions that maximize angular distortion
- Density and specific heat substantially influenced angular distortion under all heat input conditions. Thermal conductivity slightly affected angular distortion under all heat input conditions.
- Although thermal expansion coefficient largely affected angular distortion under subthreshold heat input conditions, it had less effect on angular distortion under the suprathreshold heat input condition.
- Although yield strength and elastic modulus both only slightly affected angular distortion under subthreshold heat input conditions, they both affected angular distortion more than thermal expansion coefficient did under the suprathreshold heat input condition.
- The effect of material properties on angular distortion depends on the heat input. This is because heat input affects the difference in temperature and plastic strain in the thickness direction of the substrate.
Point 4:
In Table 5, Heat Imput Condition is expressed in kJ / mm, but in Fig. 9, it seems to be expressed in kJ / m. Is it a typo? Which is correct?
Response 4:
The unit of kJ/mm for heat input is correct. Table 5 was revised.
Point 5:
There are several typos across the text, specially unexpected spaces at the beginning of paragraphs. Please check the text carefully
Response 5:
Several typos and sentences were checked and revised to make the understanding better.
Reviewer 2 Report
The paper is well written and highly interesting. Nevertheless, some improvements can be done.
The discussion is mainly referred to the simulation. The simulation was validated by the angual distortion. Later, the results are discussed for example with respect to the temperature. From a scientific point of view, a validation would be useful and a discussion under which boundary conditions the simulation is valid. Especially that these results can be used to explain the effects on other material properties.
You explain that there is a threshold at the density ration of 0.64 in Fig. 11 for Q2 and Q4, but the explanation for this effect is not discussed.
In line 371 you mention the effect of yield strength with regard to the plastic strain. Here, you assume a constant plastic strain for different materials. This simplification has to be discussed more in detail.
The discussion of the results in subsection 4.7-4.8 would benefit from a figure.
An extra row with the units could increase the readability of your tables.
A separate chapter 5 for the conclusion would help to highlight it.
With a figure, the order of effects (388-403) could be easier to understand.
Author Response
Point 1:
You explain that there is a threshold at the density ration of 0.64 in Fig. 11 for Q2 and Q4, but the explanation for this effect is not discussed.
Response 1:
The explanation was added in line 297 as follows.
In addition, when the heat input conditions (Q2, Q3) were less than the threshold (0.64), the angular distortion decreased rapidly when the density ratio decreased. This is because the difference in the plastic strain in the thickness direction is small due to the high thermal diffusivity.
Point 2:
In line 371 you mention the effect of yield strength with regard to the plastic strain. Here, you assume a constant plastic strain for different materials. This simplification has to be discussed more in detail.
Response 2:
We did not assume a constant plastic strain. Table 7 shows the contour of the plastic strain distribution. The black curve is equivalent to the contour line when the plastic strain is -0.002, and is for comparing the area of the plastic region under different material properties.
Point 3:
The discussion of the results in subsection 4.7-4.8 would benefit from a figure.
An extra row with the units could increase the readability of your tables.
A separate chapter 5 for the conclusion would help to highlight it.
With a figure, the order of effects (388-403) could be easier to understand.
Response 3:
Thanks for your recommendation. For the better understanding, the order of effects was newly presented in figure 17. In addition, a separate chapter 5 was added to highlight the findings of paper.
Reviewer 3 Report
Equation (1) should be reformed. It is not usual to have two same variables (h 2) from the both side of a sign of equality. Also, the electrical parameter voltage should be presented with U in order to have the difference with unit V (volts) because the autors have not used Italic fonts.
The equation should be: Qeff = η •U•I/ν [J/mm]
Author Response
Point 1:
Equation (1) should be reformed. It is not usual to have two same variables (h 2) from the both side of a sign of equality. Also, the electrical parameter voltage should be presented with U in order to have the difference with unit V (volts) because the autors have not used Italic fonts.
The equation should be: Qeff = η •U•I/ν [J/mm]
Response 1:
We defined the heat input parameter as Qeff/ h2 which is used for quantitative analysis of material effect on distortion. Therefore, we decided to leave it as it is. In addition, we replaced the italic variable v representing welding speed with u in Equation (1)